# Hepatitis C Virus Resistance-Associated Substitutions in Mexico

**DOI:** 10.3390/v17020169

**Published:** 2025-01-25

**Authors:** Alexis Jose-Abrego, Saul Laguna-Meraz, Sonia Roman, Irene M. Mariscal-Martinez, Arturo Panduro

**Affiliations:** 1Department of Genomic Medicine in Hepatology, Civil Hospital of Guadalajara, Fray Antonio Alcalde, Guadalajara 44280, Mexico; alexisjoseabiology@gmail.com (A.J.-A.); s.laguna.meraz@gmail.com (S.L.-M.); sonia.roman@academicos.udg.mx (S.R.); irene.mariscal@alumnos.udg.mx (I.M.M.-M.); 2Health Sciences Center, University of Guadalajara, Guadalajara 44340, Mexico; 3Doctoral Program Molecular Biology in Medicine, Health Sciences Center, University of Guadalajara, Guadalajara 44340, Mexico

**Keywords:** hepatitis C virus, subtype, antiviral resistances

## Abstract

Hepatitis C virus (HCV) is susceptible to resistance-associated substitutions (RASs) in the NS3, NS5A, and NS5B nonstructural genes, key targets of the direct-acting antivirals (DAAs). This study aimed to assess the prevalence and distribution of RASs across different HCV subtypes in Mexico. A Genbank dataset of 566 HCV sequences was analyzed. Most sequences were from Mexico City (49.1%, 278/566) and Jalisco (39.4%, 223/566). The NS5B region was the most sequenced (59.7%, 338/566). The most frequent HCV subtypes were 1a (44.0%, 249/566), 1b (28.6%, 162/566), 2b (9.5%, 54/566), and 3a (6.2%, 35/566). Subtypes 1a (57.4%, 128/223) and 3a (12.6%, 28/223) were significantly higher in Jalisco than in Mexico City (34.2%, 95/278 and 2.5%, 7/278), whereas subtype 1b was higher in Mexico City (34.5%, 96/278 vs. 14.8%, 33/223). Subtype 1a increased from 2019 to 2024, representing 49.4% (123/249) of all reported cases. RASs were detected in NS3 (6.7%, 1/15), NS5A (2.9%, 3/102), and NS5B (0.3%, 1/349), with the most frequent mutations being Q80K, Y93H, and S282T, respectively, and detected in subtypes 1b (*n* = 3), 1a (*n* = 1), and 2a (*n* = 1). In conclusion, Mexico’s HCV sequencing-based surveillance is limited. Subtype 1a predominated, but frequencies varied across states. The prevalence of RASs varied by gene from 0.3% to 6.7%. Establishing regional sequencing centers for NS3, NS5A, and NS5B is crucial to monitoring Mexico’s DAA-resistant mutations and HCV subtype genetic diversity.

## 1. Introduction

Hepatitis C virus (HCV) is a leading cause of cirrhosis and hepatocellular carcinoma worldwide [1]. In patients with chronic hepatitis C, the risk of advanced liver fibrosis is 4.26-fold greater than in individuals with chronic hepatitis B [2]. HCV transmission is associated with several risk factors, including intravenous drug use (IDU), prior incarceration, early onset of sexual activity (<17.5 years), blood transfusions, and tattooing [3,4]. In 2022, about 50 million people worldwide had chronic hepatitis C infection [5]. In Mexico, a cumulative incidence of 42,669 HCV cases was reported between 2001 and 2022 [6,7], showing a significant increase from 2020 to 2022 [6] due partially to improved access to serological tests promoted in Mexico’s national hepatitis C elimination plan [8]. Nonetheless, robust awareness campaigns, finer accessibility to treatment, and molecular epidemiology surveillance measures are lacking, hindering Mexico from reaching the 2030 elimination goal [9].

HCV has a positive-sense RNA genome of approximately 9.6 kb, which encodes ten proteins: three structural (core, E1, E2) and seven nonstructural (p7, NS2, NS3, NS4A, NS4B, NS5A, NS5B) proteins [10]. Among these, the NS3, NS5A, and NS5B proteins are targets for direct-acting antivirals (DAAs) [11]. NS3 acts as a serine protease, and NS5A regulates replication and viral assembly [12]. In contrast, NS5B is an RNA-dependent RNA polymerase replicating the viral RNA genome within the endoplasmic reticulum [12]. Due to its lack of proofreading activity, NS5B can introduce mutations at a rate of 3.5 × 10⁻^5^ to 1.2 × 10⁻^4^ base substitutions per site per year [13]. These genomic divergences have facilitated the classification of HCV into eight genotypes and over 100 subtypes [14]. Furthermore, specific mutations within the NS3, NS5A, and NS5B genes have been associated with decreased efficacy of DAAs [15]. Despite the clinical relevance of these mutations, few studies have examined HCV subtypes and RASs in Mexico. Thus, this study aimed to assess the prevalence and distribution of RASs across different HCV subtypes within the country.

## 2. Materials and Methods

### 2.1. Study Population and Design

A comprehensive dataset was retrieved from GenBank (https://www.ncbi.nlm.nih.gov/labs/virus/vssi/#/, accessed on 10 May 2024) to investigate the genetic diversity and antiviral resistance using the query (“*Hepacivirus hominis*” [Organism] OR hepatitis C virus [All Fields]) AND (“*Hepacivirus hominis*” [Organism] OR HCV [All Fields]). By 11 September 2024, 273,224 HCV sequences had been reported, of which 672 originated from Mexico. Redundant sequences and those labeled clones (*n* = 106) were excluded from the analysis. The final dataset included 566 sequences, from which information such as state of origin, genomic region, HCV genotype, and year of deposit in GenBank was collected (Figure 1).

### 2.2. HCV Genotyping

The genotypes reported in GenBank were individually confirmed using the NCBI Genotyping Tool (https://www.ncbi.nlm.nih.gov/projects/genotyping/formpage.cgi, accessed on 15 June 2024) and the HCV Typing Tool (https://www.genomedetective.com/app/typingtool/hcv/, accessed on 20 July 2024). An HCV genotype was assigned when both tools provided consistent results. Sequences that could not be identified or exhibited discrepancies were classified as “Not Available” (NA).

### 2.3. Detection of RASs

The RAS analysis was focused on three regions—NS3 (*n* = 15), NS5A (*n* = 102), and NS5B (*n* = 349)—using the Geno2pheno tool (https://hcv.geno2pheno.org/, accessed on 25 August 2024) [16]. This platform evaluated 460 clinically significant mutations (see Appendix A [17,18,19,20,21,22,23,24,25,26,27,28,29,30,31,32,33,34,35,36,37,38,39,40,41,42,43,44,45,46,47,48,49,50,51,52,53,54,55,56,57,58,59,60,61,62,63,64,65,66,67,68,69,70,71,72,73,74,75,76,77,78,79,80,81,82,83,84,85,86,87,88,89,90,91,92,93,94,95,96,97,98,99,100,101,102,103,104,105,106,107,108,109,110,111]) and classified them into four categories: resistant, reduced susceptibility, substitution on scored position, and susceptible. RASs were defined as resistant mutations with published evidence linking them to therapeutic failure. As reported in the literature, reduced susceptibility mutations were associated with decreased drug efficacy but did not wholly compromise treatment effectiveness. Substitutions at scored positions referred to mutations at critical sites that may influence drug resistance scores, although they lacked supporting evidence in the literature. Susceptible mutations were identified as not affecting drug efficacy [16].

RASs in the NS3 region associated with resistance to Asunaprevir, Boceprevir, Glecaprevir, Grazoprevir, Paritaprevir, Simeprevir, Telaprevir, and Voxilaprevir were analyzed. In the NS5A region, RASs linked to Daclatasvir, Ledipasvir, Ombitasvir, Velpatasvir, Pibrentasvir, Elbasvir, Sofosbuvir, and Dasabuvir were studied. In contrast, RASs related to Sofosbuvir and Dasabuvir in the NS5B region were investigated. RASs identified by Geno2pheno were manually confirmed by constructing datasets for each genomic region. Sequences were aligned using MEGA v11.0.13 [112] with the ClustalW algorithm, and protein sequences were translated based on the open reading frames (ORFs) for each region. For NS3, the ORF was defined by the APITAYA origin motif and LETTMR termination motif, resulting in a protein of 181 amino acids (aa). For NS5A, the ORF was defined by the SGSWL origin motif and HITAET termination motif (213aa). For NS5B, the ORF was defined by the SMSYSW origin motif and FLLPAR* termination motif (* indicating a stop codon) (292aa).

### 2.4. Statistical Analysis

Descriptive statistics were used to summarize the distribution of HCV genotypes, RAS categories, and demographic characteristics. Categorical data were expressed as frequencies or percentages, and comparisons between groups were conducted using chi-square or Fisher’s exact test, depending on the data characteristics and sample size. A *p*-value of <0.05 was considered statistically significant for all tests. Statistical analyses were performed using SPSS version 21 (IBM Corp., Armonk, NY, USA).

## 3. Results

### 3.1. Hepatitis C Sequences in Mexico

At the time of this study, 566 HCV sequences from several regions of Mexico were detected in GenBank. Most sequences were from Mexico City (49.1%, 278/566) and Jalisco (39.4%, 223/566), followed by Puebla (9.0%, 51/566), Nuevo León (1.8%, 10/566), San Luis Potosí (0.5%, 3/566), and Zacatecas (0.2%, 1/566) (Figure 2A). The NS5B region was the most sequenced (59.7%, 338/566) (Figure 2B). Mexico City reported the highest diversity of HCV sequences, including NS5B (41.4%, 115/278), NS5A (27.7%, 77/278), E1 (9.7%, 27/278), 5′ UTR (5.4%, 15/278), NS4B/NS5A (5.0%, 14/278), NS4A (4.7%, 13/278), Core (4.0%, 11/278), NS3 (1.8%, 5/278), and NS5A/NS5B (0.4%, 1/278). In contrast, Jalisco reported exclusively the NS5B region (*n* = 223), while Puebla (*n* = 51), San Luis Potosí (*n* = 3), and Zacatecas (*n* = 1) focused on the Core region. Quasi-complete genomes (Q-CG) (*n* = 10) were reported exclusively by the state of Nuevo León.

### 3.2. Hepatitis C Subtypes and Their Trends

Overall, the most common HCV subtype was 1a (44.0%, 249/566), followed by 1b (28.6%, 162/566), 2b (9.5%, 54/566), 3a (6.2%, 35/566), 4d (4.8%, 27/566), 2j (2.1%, 12/566), 2a (1.8%, 10/566), 2k/2m (0.2%, 1/566), and 2r (0.2%, 1/566) (Figure 2C). In all 5′ UTR sequences (*n* = 15), the HCV subtype could not be confirmed and was classified as “Not Available” (NA).

The distribution of HCV subtypes showed geographical variation across states. Subtypes 1a and 3a were significantly more prevalent in Jalisco compared to Mexico City (1a: 57.4%, 128/223 vs. 34.2%, 95/278, *p* = 3.25 × 10^−7^; 3a: 12.6%, 28/223 vs. 2.5%, 7/278, *p* = 2.62 × 10^−5^). In contrast, subtype 1b was more common in Mexico City than in Jalisco (1b: 34.5%, 96/278 vs. 14.8%, 33/223, *p* = 8.757 × 10^−7^). There were no significant differences in the distribution of subtypes 2a and 2b between the two states (2a: 1.1%, 3/278 in Mexico City vs. 3.1%, 7/223 in Jalisco, *p* = 0.118; 2b: 8.6%, 24/278 in Mexico City vs. 12.1%, 27/223 in Jalisco, *p* = 0.258). Subtypes 2k/2m, 2r, and 4d were only reported in Mexico City. Notably, Nuevo León, despite having fewer sequences, showed a remarkable concentration of subtype 1b compared to 1a (80.0%, 8/10 vs. 20%, 2/10, *p* = 0.023). In Puebla, subtype 1b tended to be more frequent than subtype 1a (1b: 49.0%, 25/51 vs. 1a: 39.2%, 20/51; *p* = 0.425), with minimal cases of 2b (5.9%, 3/51) and 2j (2.0%, 1/51). With minimal sequence representation, San Luis Potosí (*n* = 3) and Zacatecas (*n* = 1) displayed 100% subtype 1a.

Between 2010 and 2024, the analysis showed shifts in HCV subtype prevalence (Figure 2D). Subtypes 2r (1 case) and 2k/2m (1 case) were detected exclusively in 2018, while subtype 4d (27 cases) was first identified in 2022. None of these subtypes were detected in subsequent years. From 2010 to 2024, subtype 2a (from 2 to 5 cases) and subtype 3a (from 3 to 25 cases) showed an upward trend, whereas subtype 2j declined from 6 to 1 case. The trends for subtypes 1b and 2b were relatively more irregular. Subtype 1b ranged from 16 cases in 2010 to 38 cases in 2024, with a peak of 36 cases in 2019, before falling to 7 cases in 2023. Similarly, subtype 2b fluctuated, increasing from 5 cases in 2010 to 25 cases in 2024. Notably, subtype 1a experienced a significant increase from 36 cases in 2019 to 123 cases in 2024, accounting for nearly half of all reported cases (49.4%, 123/249), signaling its growing dominance.

### 3.3. RASs in the NS3 Gene

Based on the NS3 region, the prevalence of resistance mutations was 6.7% (1/15), while reduced susceptibility mutations accounted for 6.7% (1/15) (Table 1). The most frequent resistance mutation was 80K (*n* = 1), which confers resistance to Simeprevir. The 80R mutation was also detected in one case and was associated with reduced susceptibility to Simeprevir and Voxilaprevir.

Substitution analysis at scored positions revealed distinct mutation patterns across different antiviral agents. Mutations associated with resistance to Boceprevir and Telaprevir exhibited the highest frequency, with 53.3% (8/15) of sequences showing variations, including 170I (*n* = 5), 170I + 174T (*n* = 1), 174C (*n* = 1), and 174N (*n* = 1). For Voxilaprevir, resistance-associated mutations were observed at 122G + 170I (*n* = 5), 122T *n*+ 170I (*n* = 1), and 80R (*n* = 1), occurring in 46.7% (7/15) of sequences. The frequency of resistance-associated substitutions at scored positions for Simeprevir was 40.0% (6/15), specifically 122T + 170I. No resistance-associated or reduced susceptibility mutations were detected for Asunaprevir, Glecaprevir, Grazoprevir, or Paritaprevir.

### 3.4. RASs in the NS5A Gene

Between 2.0% (2/102) and 2.9% (3/102) of NS5A sequences had resistance-associated mutations. The most common resistance-associated mutation was 93H, known to confer resistance to multiple antivirals, including Daclatasvir (*n* = 2), Ledipasvir (*n* = 2), Ombitasvir (*n* = 2), Elbasvir (*n* = 2), and Velpatasvir (*n* = 2) (Table 2). In addition, one sequence had the 31F mutation associated with resistance to Elbasvir. Reduced susceptibility mutations were identified in 5.9% (6/102) of sequences for Velpatasvir and 4.9% (5/102) for Ombitasvir. These mutations included 28V (*n* = 4), 28V + 58S (*n* = 1), and 31F (*n* = 1) for Velpatasvir and 28V (*n* = 5) for Ombitasvir.

Analysis of substitutions at scored positions revealed frequencies ranging from 1.0% (1/102) to 10.8% (11/102). The highest frequency was observed for Ledipasvir (10.8%, 11/102), with resistance-associated mutations including 28V (*n* = 4), 28V + 58S (*n* = 1), 58R (*n* = 1), 58P (*n* = 1), 58S (*n* = 1), 58L (*n* = 1), 31F (*n* = 1), and 31L (*n* = 1). Substitutions associated with Daclatasvir were present in 9.8% (10/102) of sequences, featuring mutations such as 28V (*n* = 4), 28V + 58S (*n* = 1), 58L (*n* = 1), 58R (*n* = 1), 58P (*n* = 1), 31F (*n* = 1), and 31L (*n* = 1). Substitutions associated with Elbasvir were detected in 7.8% (8/102) of sequences, with mutations including 28V (*n* = 4), 28V + 58S (*n* = 1), 58L (*n* = 1), 58R (*n* = 1), and 58P (*n* = 1). Substitution frequencies for both Ombitasvir and Pibrentasvir were 3.9% (4/102). For Ombitasvir, the mutations observed in NS5A sequences were 58L (*n* = 1), 58R (*n* = 1), 58P (*n* = 1), and 31L (*n* = 1). Similarly, Pibrentasvir-associated substitutions included 58L (*n* = 1), 58S (*n* = 1), 58R (*n* = 1), and 58P (*n* = 1). Velpatasvir exhibited the lowest frequency of substitutions, identified in 1.0% (1/102) of sequences, represented by the 31F mutation.

### 3.5. RASs in the NS5B Gene

For Sofosbuvir, the prevalence of resistance-associated mutations and reduced susceptibility in the NS5B region was 0.3% (1/349) and 1.1% (4/349), respectively. The identified resistance-associated mutation was 282T, while mutations associated with reduced susceptibility included 289L (*n* = 3) and 321L (*n* = 1). Substitutions at scored positions were detected in 5.4% (19/349) of sequences, with 321F (*n* = 11) being the most common, followed by 321I (*n* = 2), 321T (*n* = 1), 282A (*n* = 1), 289I (*n* = 1), 282L + 321H (*n* = 1), and 282A + 320F + 321T (*n* = 1). For Dasabuvir, substitutions at scored positions were detected in 2.0% (7/349) of viral sequences, specifically 316W (*n* = 4), 316L (*n* = 1), 444D (*n* = 1), and 556I + 557W (*n* = 1) (Table 3).

### 3.6. Drug Resistances by HCV Genotypes

Of the nine HCV subtypes identified in this study, resistance-associated or reduced susceptibility mutations were only detected in HCV subtypes 1a, 1b, and 2a (Figure 3A–C). In the NS3 region, subtype 1a had resistance to Simeprevir (*n* = 1) and reduced susceptibility to Voxilaprevir (*n* = 1) and Simeprevir (*n* = 1). The NS5A region, particularly subtype 1b, exhibited the highest resistance to multiple drugs, including Elbasvir (*n* = 3), Daclatasvir (*n* = 2), Ledipasvir (*n* = 2), and Velpatasvir (*n* = 2). Subtype 1a demonstrated reduced susceptibility to Ombitasvir (*n* = 5) and Velpatasvir (*n* = 5). Finally, the NS5B region revealed reduced susceptibility (*n* = 3) and resistance (*n* = 1) to Sofosbuvir in subtype 2a, with one case of reduced susceptibility in subtype 1a (*n* = 1).

## 4. Discussion

This study reported HCV sequences from GenBank in 6 of Mexico’s 32 states: Mexico City, Jalisco, Puebla, Nuevo León, San Luis Potosí, and Zacatecas. Nearly 80% of the sequences originated from Mexico City and Jalisco. In contrast to serological studies, sequencing-based HCV surveillance in Mexico remains notably limited. An analysis of patients with social security identified HCV cases across all states in the country, with the highest prevalence in the State of Mexico (45%), followed by Mexico City (26%), Guanajuato (12%), Chihuahua (5%), Baja California (4%), Aguascalientes (2%), and Jalisco (1%) [18]. Another report on national HCV incidence found that the highest number of cases occurred in Baja California (19%), followed by Mexico City (11%), Jalisco (10%), Sinaloa (7%), the State of Mexico (7%), and Chihuahua (5%) [113]. We identified three critical epidemiological patterns by analyzing the data derived from sequencing and combined with social security records and national HCV incidence statistics. First, a high concentration of HCV cases in border states, such as Baja California and Chihuahua, suggests cross-border transmission dynamics, likely driven by migration patterns and healthcare disparities [103,113,114]. Second, a “hepatitis C bridge” was observed between the states of Jalisco, Guanajuato, the State of Mexico, and Mexico City, where HCV prevalence and incidence rates showed significant overlap, indicating shared transmission pathways or high-risk practices [114]. Finally, an emerging prevalence of HCV in tourist hubs, such as Rosarito Beach, Baja California, Cancun, Quintana Roo, and Acapulco, Guerrero, highlights the potential role of population mobility and tourism in facilitating HCV transmission [18]. In these areas, the use of intravenous psychoactive substances was strongly associated with increased anti-HCV positivity among men [113]. These findings highlight the need to improve hepatitis C surveillance in all states of Mexico, with special emphasis on other tourist hubs, such as Puerto Escondido, Oaxaca (southwest coast), Puerto Vallarta, Jalisco (central-west coast), Tampico, Tamaulipas (northeast coast), and the border city of Reynosa in Tamaulipas (north), where epidemiological data on HCV are currently missing.

Our study identified HCV subtypes 1a, 1b, 2b, and 3a as the most prevalent in Mexico, consistent with previous reports [114,115]. Globally, genotype 1 is recognized as the most widespread [116]. We observed that subtypes 1a and 3a were more common in Jalisco, while subtype 1b was significantly more prevalent in Mexico City. Another study has reported a similar trend [117]. These regional variations likely reflect the differences in transmission dynamics and risk factors unique to each region. In Jalisco, subtypes 1a and 3a have been associated with high-risk behaviors, such as intravenous drug use and tattoos, as well as with vulnerable populations, including prison inmates and patients with human immunodeficiency virus (HIV) [2,3]. In contrast, subtype 1b, more frequent in Mexico City, is predominantly linked to medical procedures, such as blood transfusions, surgery, and hemodialysis [3]. We also identified a recombinant sequence with the 2k/2m pattern, consistent with previous studies that have detected similar recombinants using a line probe assay (LiPA), including 2k/2j, 2a/1a, 2b/1b, 2a/2c, 2a/2c/4e, and 2a/2c/1a [118]. HCV recombination occurs during viral replication when the RNA-dependent RNA polymerase (RdRp) switches between distinct viral RNA templates, generating a recombinant genome [119]. This process is a natural mechanism for enhancing genetic diversity, which is critical for the virus’s ability to adapt [120]. Recombination facilitates rapid evolution, driven by selective pressures, such as immune responses or antiviral treatment [10]. Another rare subtype was 4d, identified in a population of HIV patients [121]. These isolates showed a close phylogenetic relationship with strains from France [121]. Similar outbreaks of subtype 4d have been reported in Quebec, Canada, and Paris, France [122,123]. These rare subtypes have been exclusively identified in Mexico City, likely due to its status as the nation’s capital and a major destination for foreign tourists, facilitating the introduction of novel subtypes [123].

Since their introduction in 2017, DAAs have significantly improved the treatment of HCV infection in Mexico [8]. A meta-analysis revealed that 78–93% of patients who fail to achieve a cure with DAAs harbor RASs [124]. In this study, RASs were detected in the NS3 (6.7%), NS5A (2.9%), and NS5B (0.3%) regions, with notable mutations including Q80K in NS3, Y93H in NS5A, and S282T in NS5B, predominantly in subtypes 1b, 1a, and 2a. A previous study focusing exclusively on the NS5A region reported an RAS prevalence of 29% in Mexico, which was primarily associated with subtype 1b [125]. A meta-analysis revealed that the prevalence of RASs in the NS3, NS5A, and NS5B genes varies across geographic regions [32]. The highest RAS burden was observed in Thailand and Japan, while a high–intermediate burden was reported in the United States, the United Kingdom, and Germany. An intermediate RAS burden was identified in Canada, China, and Australia, whereas countries such as Spain, France, Denmark, Swaziland, New Zealand, Vietnam, and Pakistan exhibited an intermediate–low burden. In contrast, Brazil, Egypt, Russia, India, and Italy had a low RAS burden [126]. The report also indicated that subtypes 1b, 1a, 6, and 3 are susceptible to RASs in the NS3 gene, while subtypes 1b, 6, 3, and 4 exhibit a higher prevalence of RASs in the NS5A gene. For the NS5B gene, RASs are most frequently found in subtypes 1b, 4, and 2 [126]. In North America, subtype 1a predominates, and specific NS5A RASs, such as M28V and A92T, have been associated with resistance to inhibitors like Ledipasvir, Velpatasvir, and Ombitasvir [16]. These RASs have been previously reported in Mexico in 9% and 1.8% of cases, respectively [125]. In our study, we found five patients (4.9%) with the M28V RAS, highlighting the need for ongoing research into multi-resistant mutations to better understand their dynamics, particularly in the era of DAA therapies.

Although the prevalence of RASs in this study is relatively low, these mutations are clinically significant, particularly in patients with limited treatment options. Subtype 1b, often associated with accelerated liver disease progression, has the highest number of RASs globally. In Mexico, subtype 1b is the second most prevalent genotype after subtype 1a, contributing to a substantial proportion of resistance mutations. For example, combinations like F37L + Q54H were observed in 12.5% of subtype 1b cases, reducing the efficacy of multiple NS5A inhibitors [125]. Differences in RAS prevalence may result from various factors, including differential selective pressures exerted by antiviral treatments, variability in sample sizes, intrinsic genetic differences between HCV subtypes, or disparities in treatment access [127]. Additionally, patients previously treated with less effective medications may have developed resistant variants, and co-infections with other viruses, such as HIV, could alter viral dynamics, facilitating the emergence of resistance mutations due to changes in the immunological or pharmacological environment [128].

Our findings align with global trends, showing that RASs occur at relatively low frequencies in Mexico. However, identifying key mutations, such as Q80K, Y93H, and S282T, highlights the need for ongoing surveillance. Considering the widespread use of DAAs, including Epclusa^®^ (Sofosbuvir/Velpatasvir) and Mavyret^®^ (Glecaprevir/Pibrentasvir), proactive monitoring of resistance patterns is crucial to ensure the continued efficacy of these therapies in managing HCV infections in Mexico. While this study offers valuable insights into HCV resistance mutations, its small sample size, limited data on the NS3 and NS5A regions, and the restricted geographic scope—covering sequences from only two states—limit the generalizability of the findings to the entire Mexican population. Despite these limitations, the study has significant strengths. It is the first study to investigate HCV resistance mutations across the three primary DAA target regions while also identifying emerging epidemiological patterns of HCV in Mexico. These findings lay the groundwork for future research, underscoring the need for larger, more geographically representative samples and improved sequencing coverage of underrepresented regions like NS3 and NS5A. Such efforts will be essential for accurately monitoring resistance mutation prevalence and assessing the effectiveness of antiviral therapies across Mexico.

## 5. Conclusions

In conclusion, HCV surveillance through sequencing is limited in Mexico. Subtype 1a was the most prevalent, though its frequency varied by state. The prevalence of HCV resistance mutations ranged from 0.3% to 6.7%, depending on the gene analyzed. Establishing regional sequencing centers for NS3, NS5A, and NS5B sequencing is essential for tracking potential resistance mutations to DAAs and evaluating the evolving genetic diversity of HCV subtypes in Mexico.

## Figures and Tables

**Figure 1 viruses-17-00169-f001:**
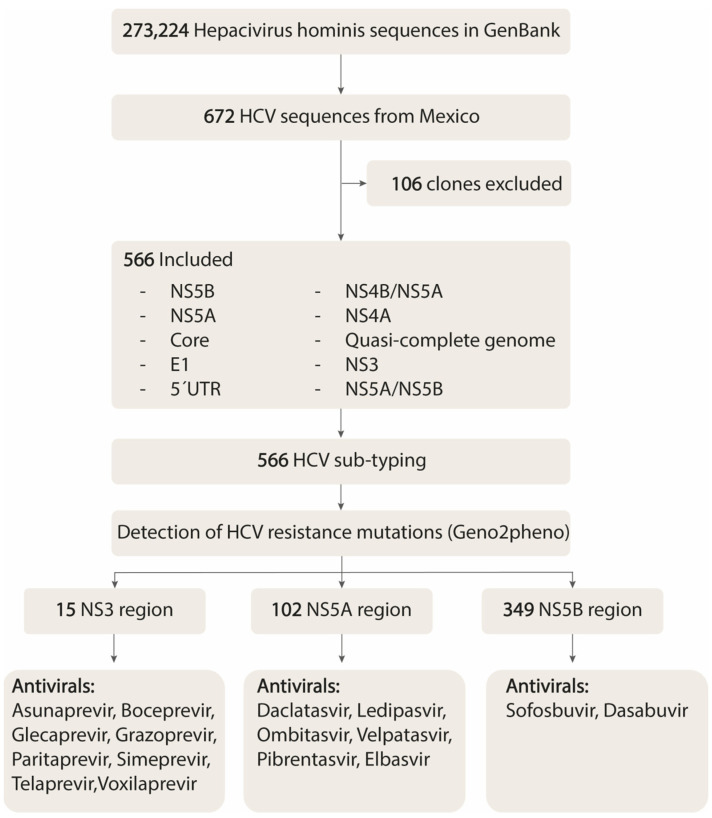
Data collection strategy for analyzing HCV genotypes and antiviral resistance in Mexico.

**Figure 2 viruses-17-00169-f002:**
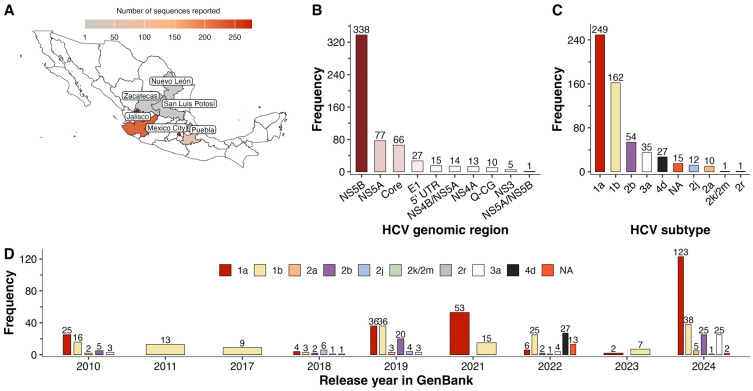
Epidemiology of HCV in Mexico based on GenBank sequences. Geographic distribution of reported HCV sequences (**A**). Frequency of HCV genomic regions sequenced (**B**) and distribution of HCV subtypes (**C**). Temporal trends in HCV subtype prevalence from 2010 to 2024 (**D**). Q-CG: Quasi-complete genome, NA: Not available, 2k/2m refers to the recombinant HCV subtype.

**Figure 3 viruses-17-00169-f003:**
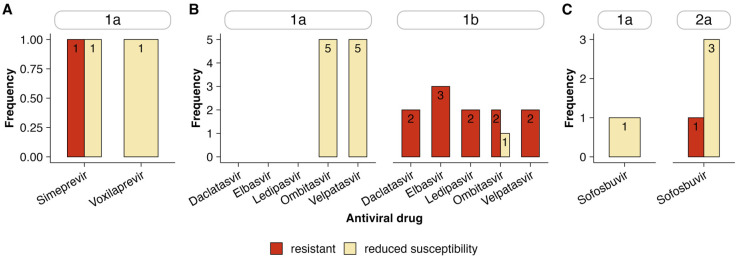
Frequency of resistance and reduced susceptibility mutations to antiviral drugs by HCV subtype based on NS3 (**A**), NS5A (**B**), and NS5B (**C**) regions.

**Table 1 viruses-17-00169-t001:** RASs in the HCV NS3 gene.

Antiviral	Category	Total (N = 15)
Simeprevir	reduced susceptibility (80R *n* = 1)	1 (6.7%)
	resistant (80K *n* = 1)	1 (6.7%)
	substitution on scored position (122T + 170I *n* = 6)	6 (40.0%)
	susceptible	7 (46.7%)
Voxilaprevir	reduced susceptibility (80K *n* = 1)	1 (6.7%)
	substitution on scored position (122G + 170I *n* = 5, 122T + 170I *n* = 1, 80R *n* = 1)	7 (46.7%)
	susceptible	7 (46.7%)
Boceprevir	substitution on scored position (170I *n* = 5, 170I + 174T *n* = 1, 174C *n* = 1, 174N *n* = 1)	8 (53.3%)
	susceptible	7 (46.7%)
Telaprevir	substitution on scored position (170I *n* = 5, 170I + 174T *n* = 1, 174C *n* = 1, 174N *n* = 1)	8 (53.3%)
	susceptible	7 (46.7%)
Asunaprevir	susceptible	15 (100.0%)
Glecaprevir	susceptible	15 (100.0%)
Grazoprevir	susceptible	15 (100.0%)
Paritaprevir	susceptible	15 (100.0%)

Quasi-complete genome *n* = 10; NS3 *n* = 5. HCV: hepatitis C virus.

**Table 2 viruses-17-00169-t002:** RASs in the HCV NS5A gene.

Antiviral	Category	Total (N = 102)
Daclatasvir	resistant (93H *n* = 2)	2 (2.0%)
	substitution on scored position (28V *n* = 4, 28V + 58S *n* = 1, 58L *n* = 1, 58R *n* = 1, 58P *n* = 1, 31F *n* = 1, 31L *n* = 1)	10 (9.8%)
	susceptible	90 (88.2%)
Ledipasvir	resistant (93H *n* = 2)	2 (2.0%)
	substitution on scored position (28V *n* = 4, 28V + 58S *n* = 1, 58R *n* = 1, 58P *n* = 1, 58S *n* = 1, 58L *n* = 1, 31F *n* = 1, 31L *n* = 1)	11 (10.8%)
	susceptible	89 (87.3%)
Ombitasvir	reduced susceptibility (28V *n* = 4, 28V + 58S *n* = 1, 31F *n* = 1)	6 (5.9%)
	resistant (93H *n* = 2)	2 (2.0%)
	substitution on scored position (58L *n* = 1, 58R *n* = 1, 58P *n* = 1, 31L *n* = 1)	4 (3.9%)
	susceptible	90 (88.2%)
Elbasvir	resistant (93H *n* = 2, 31F *n* = 1)	3 (2.9%)
	substitution on scored position (28V *n* = 4, 28V + 58S *n* = 1, 58L *n* = 1, 58R *n* = 1, 58P *n* = 1)	8 (7.8%)
	susceptible	91 (89.2%)
Velpatasvir	reduced susceptibility (28V *n* = 5)	5 (4.9%)
	resistant (93H *n* = 2)	2 (2.0%)
	substitution on scored position (31F *n* = 1)	1 (1.0%)
	susceptible	94 (92.2%)
Pibrentasvir	substitution on scored position (58L *n* = 1, 58S *n* = 1, 58R *n* = 1, 58P *n* = 1)	4 (3.9%)
	susceptible	98 (96.1%)

Quasi-complete genome *n* = 10; NS4B/NS5A *n* = 14; NS5A *n* = 77; NS5A/NS5B *n* = 1.

**Table 3 viruses-17-00169-t003:** RASs in the NS5B gene.

Antiviral	Category	Total (N = 349)
Sofosbuvir	reduced susceptibility (289L *n*= 3, 321L *n* = 1)	4 (1.1%)
	resistant (282T *n* = 1)	1 (0.3%)
	substitution on scored position (321F *n* = 11, 321I *n* = 2, 321L *n* = 1, 321T *n* = 1, 282A *n* = 1, 289I *n* = 1, 282L + 321H *n* = 1, 282A + 320F + 321T *n* = 1)	19 (5.4%)
	susceptible	325 (93.1%)
Dasabuvir	insufficient coverage	110 (31.5%)
	substitution on scored position (316W *n* = 4, 316L *n* = 1, 444D *n* = 1, 556I + 557W *n* = 1)	7 (2.0%)
	susceptible	232 (66.5%)

Quasi-complete genome *n* = 10; NS5A/NS5B *n* = 1; NS5B *n* = 338.

## Data Availability

Data contained within the article are publicly available at NCBIvirus (https://www.ncbi.nlm.nih.gov/labs/virus/vssi/#/, accessed on 10 May 2024).

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
