# Peer review of "Hepatitis C Virus Resistance-Associated Substitutions in Mexico"

_viruses, 2025, doi:10.3390/v17020169_

Round 1

Reviewer 1 Report

Comments and Suggestions for Authors

I like the paper; actually data on HCV DAA resistance in mexico are already present in other publications but not in the form of an extensive review like this one, that also complete a geo-social study of HCV and HCV subtypes in Mexico.

Surely the data would be even more interesting if the authors have available data on eventual previous treatment in the patients studied: probably they will have a group of patients naive, a group of treated and a group of unknown. Analyse these data, if possible, could be interesting

Author Response

Dear Reviewer 1

We greatly appreciate your thorough review of our manuscript "Hepatitis C Virus resistance-associated substitutions in Mexico". Your comments have provided valuable feedback that will undoubtedly enhance the quality and clarity of our work. We have carefully addressed each of your concerns and suggestions, and we present our responses below:

Q1.Regarding the comment: “I like the paper; actually data on HCV DAA resistance in Mexico are already present in other publications but not in the form of an extensive review like this one, that also complete a geo-social study of HCV and HCV subtypes in Mexico.”

A1.We value your comments on our manuscript. In structuring it, we sought to make the introduction and methodology as comprehensive as possible, ensuring a foundation for analysis. Also, we ensured that our conclusions were supported by the results. We hope that this approach provides an overview of the unique aspects of HCV resistance and the geosocial dynamics of HCV subtypes in Mexico.

Q2.Regarding the comment: “Surely the data would be even more interesting if the authors have available data on eventual previous treatment in the patients studied: probably they will have a group of patients naive, a group of treated and a group of unknown. Analyse these data, if possible, could be interesting”.

A2.We agree that comparing patients based on their previous treatment status—naïve, treated, or unknown—would add significant clinical relevance to the study. However, during the data collection process, we encountered a limitation: very few of the sequences included information on treatment status, including whether the patients were solely HCV-infected or coinfected with HIV. As a result, we were unable to conduct the proposed analysis.

We appreciate your time in reviewing our manuscript.

Reviewer 2 Report

Comments and Suggestions for Authors

In this manuscript, AlexisJose-Abrego examined the presence of antiviral resisitance mutations in hepatitis C virus sequences gathered from NCBI data. Overall, the authors discovered a low rate of resistance mutations although NS5A of subtype 1b seems to be affected by an already important diversity of mutants in Mexico. The authors insist on the necessity to widen such exploration to all 32 mexican states (only six of them essentially located around Mexico city were represented in the current study).

The paper is sound and well-written. As emphasized by the authors, It brings some important information on a large country as Mexico that can be the cradle of various types of hepatitis C epidemics either linked to intravenous drugs consumption or to mass tourism.

There are only few minor issues with the manuscript.

NS5A mutations are mentioning Croatia and then Europe. But Croatia ins in Europe.Please rephrase.

"Boceprevir and Telaprevir exhibited the highest frequency of mutations": All the manuscript uses this colloquial approximation such as "boceprevir mutations", etc... I do not think we shoudl use this confusing description as it is not boceprevir but the viral genome that is mutated. In my opinion, we should privilege, even if it is heavier, "boceprevir-associated mutations".

The rate of mutation mentioned in the discussion are also confusing eg:

"highest prevalence of NS5B RASs was observed in Taiwan (34.3%) and Australia (28.2%)": does it mean that 34.3% of HCV strains from Taiwan present a RAS in NS5B? It sounds huge and it means that sooner or later, NS5B inhibitors will not be used anymore in Taiwan. All that paragraph is a bit like that. Please, clarify.

Author Response

Dear Reviewer 2

We greatly appreciate your thorough review of our manuscript "Hepatitis C Virus resistance-associated substitutions in Mexico". Your comments have provided valuable feedback that will undoubtedly enhance the quality and clarity of our work. We have carefully addressed each of your concerns and suggestions, and we present our responses below:

Q1.Regarding the comment: “NS5A mutations are mentioning Croatia and then Europe. But Croatia ins in Europe. Please rephrase.”.

A1. We apologize for the lack of clarity in our original statement. We intended to highlight the differences in RAS burdens among various countries and emphasize the HCV genotypes with higher susceptibility to RAS. We have rewritten this section in the revised manuscript for improved clarity. Please refer to page 9, lines 268–283, in the updated version.

Q2.Regarding the comment: “Boceprevir and Telaprevir exhibited the highest frequency of mutations": All the manuscript uses this colloquial approximation such as "boceprevir mutations", etc... I do not think we shoudl use this confusing description as it is not boceprevir but the viral genome that is mutated. In my opinion, we should privilege, even if it is heavier, "boceprevir-associated mutations.".

A2. We have implemented your recommendation in the document. Please see the sentences highlighted in yellow in the updated version.

Q3. Regarding the comment: “highest prevalence of NS5B RASs was observed in Taiwan (34.3%) and Australia (28.2%)": does it mean that 34.3% of HCV strains from Taiwan present a RAS in NS5B? It sounds huge and it means that sooner or later, NS5B inhibitors will not be used anymore in Taiwan. All that paragraph is a bit like that. Please, clarify.”

A3. We understand that this sentence may have caused some confusion. This paragraph has been improved and updated. Please refer to page 9, lines 268–283 in the updated version.

We appreciate your attention to detail and have taken your suggestions into account to enhance the clarity and effectiveness of our manuscript.